# A Rotation and a Translation Suffice: Fooling CNNs with Simple Transformations

## Abstract

We show that simple spatial transformations, namely translations and rotations alone, suffice to fool neural networks on a significant fraction of their inputs in multiple image classification tasks. Our results are in sharp contrast to previous work in adversarial robustness that relied on more complicated optimization approaches unlikely to appear outside a truly adversarial context. Moreover, the misclassifying rotations and translations are easy to find and require only a few black-box queries to the target model. Overall, our findings emphasize the need to design robust classifiers even for natural input transformations in benign settings.

## 1 Introduction

Neural networks are now widely embraced as dominant solutions in computer vision (Krizhevsky et al., 2012; He et al., 2016), speech recognition (Graves et al., 2013), and natural language processing (Collobert & Weston, 2008). While their accuracy scores often match (and sometimes go beyond) human-level performance on key benchmarks (He et al., 2015; Taigman et al., 2014), we still do not understand how robust neural networks are. A prominent issue in this context is the existence of so-called *adversarial examples*, i.e., inputs that are almost indistinguishable from natural data to a human but cause state-of-the-art classifiers to make incorrect predictions with high confidence (Szegedy et al., 2013; Goodfellow et al., 2014). This raises concerns about the use of neural networks in contexts where reliability, dependability, and security are important desiderata.

There is a long line of work on methods for constructing adversarial perturbations in various settings (Szegedy et al., 2013; Goodfellow et al., 2014; Kurakin et al., 2016a;b; Sharif et al., 2016; Moosavi-Dezfooli et al., 2016; Carlini & Wagner, 2016; Papernot et al., 2017; Madry et al., 2017; Athalye et al., 2017). However, these methods are quite sophisticated and the resulting perturbations tend to be fairly contrived since they often rely on fine-tuned control over a large number of input pixels or audio samples. So one may suspect that adversarial examples constitute a problem only in the presence of a truly malicious attacker and are unlikely to arise in more benign environments. In particular, the focus on intricate worst-case attacks so far raises a natural question:

*Are neural networks robust to simple, naturally-occurring transformations of their input?*

We address this question by studying two basic image transformations: translations and rotations. While these transformations appear natural to a human, we show that small rotations and translations *alone* (i.e., without any additional fine-tuned perturbation) can cause a significant drop in the model's performance. This holds even when the model has been trained using appropriate data augmentation and no visual information is lost due to these transformations (e.g. due to cropping, see Figure 1).

### 1.1 Our Methodology and Results

We start with standard image classifiers for the MNIST (LeCun et al., 1998), CIFAR10 (Krizhevsky & Hinton, 2009), and ImageNet (Russakovsky et al., 2015) datasets. The classifiers achieve close to state-of-the-art performance on the respective benchmarks. Nevertheless, we demonstrate that small transformations can cause a significant drop in classification accuracy for these models. Depending on dataset and model, this drop ranges from 34% to as high as 90% for the worst combination of rotation angle and translation shift. Even for a small *random* transformation, the accuracy can drop

Natural  Adversarial  Natural  Adversarial  Natural  Adversarial

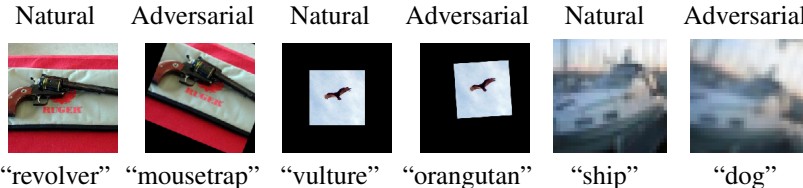

"revolver"  "mousetrap"  "vulture"  "orangutan"  "ship"  "dog"

Figure 1: Examples of adversarial transformations and their predictions in the standard, "black canvas", and reflection padding setting.

by up to 30%. These results demonstrate that robustness to rotations and translations should also be a concern in standard classification problems outside an adversarial security context.

Moreover, we show that direct access to the model (or a surrogate) is not necessary to find such misclassifying transformations. Choosing the worst out of 10 random transformations suffices to reduce the accuracy of these models by 26% on MNIST, 72% on CIFAR10, and 28% on ImageNet (top 1 accuracy). Hence our results also give a strong baseline for fooling classifiers with a small number of non-adaptive queries.

Finally, we examine possible ways to alleviate these vulnerabilities. A natural first step is to augment the training procedure with rotations and translations. While this does largely mitigate the problem on MNIST, the models trained on CIFAR10 and ImageNet are still far from robust. We thus propose two natural methods for further increasing the robustness of these models. These methods are based on robust optimization and aggregation of random input transformations. They offer significant improvements in classification accuracy but also come with considerable computational overhead. Even then, they are still not sufficient to completely mitigate the vulnerability. This suggests that obtaining models robust to spatial transformations of their inputs remains a challenge.

Finally, we examine the interplay between rotations / translations and the widely used $\ell_\infty$-based adversarial examples. We observe that robustness to these two classes of input perturbations is largely orthogonal to each other. In particular, pixel-based robustness does not imply spatial robustness, while combining spatial and $\ell_\infty$-bounded transformations seems to have a *cumulative* effect in reducing classification accuracy. This emphasizes the need to broaden the notions of image similarity in the adversarial examples literature beyond the common $\ell_p$-balls.

## 1.2 SUMMARY OF CONTRIBUTIONS

We perform extensive experiments that provide a fine-grained understanding of rotation / translation robustness on a wide spectrum of datasets and training regimes. In summary, we show that:

- A simple attack based solely on rotations and translations is effective against state-of-the-art neural networks. This holds even when the model has been trained with appropriate data augmentation and no image information is lost during the spatial transformation.

- Rotation / translation attacks are easy to execute, requiring only a few black-box queries.

- It is possible to increase a model's robustness to rotations and translations at the cost of increased training and / or inference time. However, these methods are still not sufficient to fully recover the accuracy on unmodified images.

- Robustness to $\ell_\infty$-bounded perturbations does not significantly affect spatial robustness. Instead, these two notions appear orthogonal to each other.

- First-order methods are significantly less effective for finding adversarial transformations than an exhaustive search over a fine grid of transformations. This is in stark contrast to $\ell_p$-bounded perturbrations where first-order methods have been very successful (Carlini & Wagner, 2016; Madry et al., 2017). Hence rigorous evaluation of model robustness in this spatial setting requires techniques that are different from $\ell_p$-bounded adversarial examples.

## 2 ADVERSARIAL ROTATIONS AND TRANSLATIONS

Recall that in the context of image classification, an *adversarial example* for a given input image $x$ and a classifier $C$ is an image $x'$ that satisfies two properties: (i) on the one hand, the adversarial example $x'$ causes the classifier $C$ to output a different label on $x'$ than on $x$, i.e., we have $C(x) \neq C(x')$. (ii) On the other hand, the adversarial example $x'$ is "visually similar" to $x$.

Clearly, the notion of visual similarity is not precisely defined here. In fact, providing a precise and rigorous definition is extraordinarily difficult as it would require formally capturing the notion of human perception. Consequently, previous work largely settled on the assumption that $x'$ is a valid adversarial example for $x$ if and only if $\|x - x'\|_p \leq \varepsilon$ for some $p \in [0, \infty]$ and $\varepsilon$ small enough. This convention is based on the fact that two images are indeed visually similar when they are close enough in some $\ell_p$ norm. However, the converse is not necessarily true. A small rotation or translation of an image usually appears visually similar to a human, yet can lead to a large change when measured in an $\ell_p$ norm. We aim to expand the range of similarity measures considered in the adversarial examples literature by investigating robustness to small rotations and translations.

**Attack methods.** Our first goal is to develop sufficiently strong methods for generating adversarial rotations and translations. In the context of pixel-wise $\ell_p$ perturbations, the most successful approach for constructing adversarial examples so far has been to employ optimization methods on a suitable loss function (Szegedy et al., 2013; Goodfellow et al., 2014; Carlini & Wagner, 2016). Following this approach, we parametrize our attack method with a set of tunable parameters and then optimize over these parameters. We perform this optimization in three distinct ways:

- **First-Order Method (FO):** Starting from a random choice of parameters, we iteratively take steps in the direction of the gradient of the loss function. This is the direction that locally maximizes the loss of the classifier (as a surrogate for misclassification probability). Note that unlike the $\ell_p$-norm case, we are not optimizing in the pixel space but in the latent space of rotation and translation parameters.

- **Grid Search:** We discretize the parameter space and exhaustively examine every possible parametrization of the attack to find one that causes the classifier to give a wrong prediction (if such a parametrization exists). Since our parameter space is low-dimensional enough, this method is computationally feasible (in contrast to a grid search for $\ell_p$-based adversaries).

- **Worst-of-$k$:** We randomly sample $k$ different choices of attack parameters and choose the one on which the model performs worst. As we increase $k$, this attack interpolates between a random choice and grid search.

While a first-order attack requires full knowledge of the model to compute the gradient of the loss with respect to the input, the other two attacks do not. They only require the outputs corresponding to chosen inputs, which can be done witho only query access to the target model.

Next, we need to define the exact range of attacks we want to optimize over. For the case of rotation and translation attacks, we wish to find parameters $(\delta u, \delta v, \theta)$ such that rotating the original image by $\theta$ degrees around the center and then translating it by $(\delta u, \delta v)$ pixels causes the classifier to make a wrong prediction. Formally, the pixel at position $(u, v)$ is moved to the following position (assuming the point $(0, 0)$ is the center of the image):

$$\begin{bmatrix} u' \\ v' \end{bmatrix} = \begin{bmatrix} \cos\theta & -\sin\theta \\ \sin\theta & \cos\theta \end{bmatrix} \cdot \begin{bmatrix} u \\ v \end{bmatrix} + \begin{bmatrix} \delta u \\ \delta v \end{bmatrix}.$$

We implement this transformation in a differentiable manner using the spatial transformer blocks of (Jaderberg et al., 2015). In order to handle pixels that are mapped to non-integer coordinates, the transformer units include a differentiable bilinear interpolation routine. Since our loss function is differentiable with respect to the input and the transformation is in turn differentiable with respect to its parameters, we can obtain gradients of the model's loss function w.r.t. the perturbation parameters. This enables us to apply a first-order optimization method to our problem.

By defining the spatial transformation for some $x$ as $T(x; \delta u, \delta v, \theta)$, we construct an adversarial perturbation for $x$ by solving the problem

$$\max_{\delta u, \delta v, \theta} \mathcal{L}(x', y), \quad \text{for } x' = T(x; \delta u, \delta v, \theta) , \tag{1}$$

where $\mathcal{L}$ is the loss function of the neural network[1], and $y$ is the correct label for $x$. Since this is a non-concave maximization problem, there are no guarantees for the global optimality of a general first order method.

## 3 Improving Invariance to Spatial Transformations

As we will see in Section 4, augmenting the training set with random rotations and translations does improve the robustness of the model against such random transformations. However, data augmentation does not significantly improve the robustness against worst-case attacks and sometimes leads to a drop in accuracy on unperturbed images. To address these issues, we explore two simple baselines that turn out to be surprisingly effective.

**Robust Optimization.** Instead of performing standard empirical risk minimization to train the classification model, we utilize ideas from robust optimization. Robust optimization has a rich history (Ben-Tal et al., 2009) and has recently been applied successfully in the context of defending neural networks against adversarial examples (Madry et al., 2017; Sinha et al., 2017; Raghunathan et al., 2018; Kolter & Wong, 2017). The main barrier to applying robust optimization for spatial transformations is the lack of an efficient procedure to compute the worst-case perturbation of a given example. Performing a grid search (as described in Section 2) is prohibitive as this would increase the training time by a factor close to the grid size, which can easily be a factor 100 or 1,000. Moreover, the non-convexity of the loss landscape prevents potentially more efficient first-order methods from discovering (approximately) worst-case transformations (see Section 4 for details).

Given that we cannot fully optimize over the space of translations and rotations, we instead use a coarse approximation provided by the worst-of-10 adversary (as described in Section 2). So each time we use an example during training, we first sample 10 transformations of the example uniformly at random from the space of allowed transformations. We then evaluate the model on each of these transformations and train on the one perturbation with the highest loss. This corresponds to approximately minimizing a min-max formulation of robust accuracy similar to (Madry et al., 2017). Training against such an adversary increases the overall time by a factor of roughly six.[2]

**Aggregating Random Transformations.** As Section 4 shows, the accuracy against a *random* transformation is significantly higher than the accuracy against the worst transformation in the allowed attack space. This motivates the following inference procedure: compute a (tyipcally small) number of random transformations of the input image and output the label that occurs most common in the resulting set of predictions. We constrain these random transformations to be within $5\%$ of the input image size in each translation direction and up to $15°$ of rotation. [3] The training procedure and model can remain unchanged while the inference time is increased by a small factor (equal to the number of transformations we evaluate on).

**Combining Both Methods.** The two methods outlined above are orthogonal and in some sense complementary. We can therefore combine robust training (using a worst-of-k adversary) and majority inference to further increase the robustness of our models.

## 4 Experiments

We evaluate standard image classifiers for the MNIST (LeCun et al., 1998), CIFAR10 (Krizhevsky & Hinton, 2009) and ImageNet (Russakovsky et al., 2015) datasets. In order to determine the extent to which misclassification is caused by insufficient data augmentation during training, we examine various data augmentation methods. We begin with a description of our experimental setup.

---

[1]The loss $\mathcal{L}$ of the classifier is a function from images to real numbers that expresses the performance of the network on the particular example $x$ (e.g., the cross-entropy between predicted and correct distributions).

[2]We need to perform 10 forward passes and one backwards pass instead of one forward and one backward pass required for standard training.

[3]Note that if an adversary rotates an image by $30°$ (a valid attack in our threat model), we may end up evaluating the image on rotations of up to $45°$.

**Model Architecture.**   For MNIST, we use a convolutional neural network derived from the TensorFlow Tutorial (tft). In order to obtain a fully convolutional version of the network, we replace the fully-connected layer by two convolutional layers with 128 and 256 filters each, followed by a global average pooling. For CIFAR10, we consider a standard ResNet (He et al., 2016) model with 4 groups of residual layers with filter sizes [16, 16, 32, 64] and 5 residual units each. We use standard and $\ell_\infty$-adversarially trained models similar to those studied by Madry et al. (2017).[4],[5] For ImageNet, we use a ResNet-50 (He et al., 2016) architecture implemented in the `tensorpack` repository (Wu et al., 2016). We did not modify the model architectures or training procedures.

**Attack Space.**   In order to maintain the visual similarity of images to the natural ones we restrict the space of allowed perturbations to be relatively small. We consider rotations of at most $30°$ and translations of at most (roughly) 10% percent of the image size in each direction. This corresponds to 3 pixels for MNIST (image size $28 \times 28$) and CIFAR10 (image size $32 \times 32$), and 24 pixels for ImageNet (image size $299 \times 299$). For grid search attacks, we consider 5 values per translation direction and 31 values for rotations, equally spaced. For first-order attacks, we use 200 steps of projected gradient descent of step size 0.01 times the parameter range. When rotating and translating the images, we fill the empty space with zeros (black pixels).

**Data Augmentation.**   We consider five variants of training for our models.

- Standard training: The standard training procedure for the respective model architecture.
- $\ell_\infty$-bounded adversarial training: The classifier is trained on $\ell_\infty$-bounded adversarial examples that are generated with projected gradient descent.
- No random cropping: Standard training for CIFAR-10 and ImageNet includes data augmentation via random crops. We investigate the effect of this data augmentation scheme by also training a model without random crops.
- Random rotations and translations: At each training step, we perform a uniformly random perturbation from the attack space on each training example.
- Random rotations and translations from larger intervals: As before, we perform uniformly random perturbations, but now from a *superset* of the attack space ($40°, \pm 13\%$ pixels).

### 4.1   EVALUATING MODEL ROBUSTNESS

We evaluate all models against random and grid search adversaries with rotations and translations considered both separately and together. We report the results in Table 1. We visualize a random subset of successful attacks in Figures 3, 4, and 5 of Appendix A.

Despite the high accuracy of standard models on unperturbed examples and their reasonable performance on random perturbations, a grid search can significantly lower the classifiers' accuracy on the test set. For the standard models, accuracy drops from 99% to 26% on MNIST, 93% to 3% on CIFAR10, and 76% to 31% on ImageNet (Top 1 accuracy).

The addition of random rotations and translations during training greatly improves both the random and adversarial accuracy of the classifier for MNIST and CIFAR10, but less so for ImageNet. For the first two datasets, data augmentation increases the accuracy against a grid adversary by 60% to 70%, while the same data augmentation technique adds less than 3% accuracy on ImageNet.

In Appendix A, we perform a fine-grained investigation of our findings:

- In Figure 8 we examine how many examples can be fooled by (i) rotations only, (ii) translations only, (iii) neither transformation, or (iv) both.
- We visualize the set of fooling angles for a random sample of the rotations-only grid in Figure 9. We observe that the set of fooling angles is not contiguous.
- To investigate how many transformations are adversarial per image, we analyze the percentage of misclassified grid points for each example in Figure 10. While the majority of

---

[4]`https://github.com/MadryLab/cifar10_challenge`
[5]`https://github.com/MadryLab/mnist_challenge`

images has only a small number of adversarial transformations, a significant fraction of images is fooled by 20% or more of the transformations.

Table 1: Accuracy of different classifiers against rotation and translation adversaries on MNIST, CIFAR10, and ImageNet. The allowed transformations are translations by (roughly) 10% of the image size and $\pm 30°$ rotations. The attack parameters are chosen through random sampling or grid search with rotations and translations considered both together ("Rand.", "Grid") and separately ("Rand. T." and "Grid T." for transformations, "Rand R." and "Grid R." for rotations). We consider networks that are trained with (i) the respective standard setup, (ii) no data augmentation (if data augmentation is present in standard setup), (iii) with an $\ell_\infty$ adversary, (iv) with data augmentation corresponding to the attack space ($\pm 3px$, $\pm 30°$) and an enlarged space ($\pm 4px$, $\pm 40°$), and (v) with worst-of-10 training for both types of augmentations.

|  | Model | Nat. | Rand. | Grid | Rand. T. | Grid T. | Rand. R. | Grid R. |
|---|---|---|---|---|---|---|---|---|
| MNIST | Standard | 99.31% | 94.23% | **26.02%** | 98.61% | 89.80% | 95.68% | 70.98% |
|  | $\ell_\infty$-Adv | 98.65% | 88.02% | **1.20%** | 93.72% | 34.13% | 95.27% | 72.03% |
|  | Aug. 30 | 99.53% | 99.35% | **95.79%** | 99.47% | 98.66% | 99.34% | 98.23% |
|  | Aug. 40 | 99.34% | 99.31% | **96.95%** | 99.39% | 98.65% | 99.40% | 98.49% |
|  | W-10 (30) | 99.48% | 99.37% | **97.32%** | 99.50% | 99.01% | 99.39% | 98.62% |
|  | W-10 (40) | 99.42% | 99.39% | **97.88%** | 99.45% | 98.89% | 99.36% | 98.85% |
| CIFAR10 | Standard | 92.62% | 60.93% | **2.80%** | 88.54% | 66.17% | 75.36% | 24.71% |
|  | No Crop | 90.34% | 54.64% | **1.86%** | 81.95% | 46.07% | 69.23% | 18.34% |
|  | $\ell_\infty$-Adv | 80.21% | 58.33% | **6.02%** | 78.15% | 59.02% | 62.85% | 20.98% |
|  | Aug. 30 | 90.02% | 90.92% | **58.90%** | 91.76% | 79.01% | 91.14% | 76.33% |
|  | Aug. 40 | 88.83% | 91.18% | **61.69%** | 91.53% | 77.42% | 91.10% | 76.80% |
|  | W-10 (30) | 91.34% | 92.35% | **69.17%** | 92.43% | 83.01% | 92.33% | 81.82% |
|  | W-10 (40) | 91.00% | 92.11% | **71.15%** | 92.28% | 82.15% | 92.53% | 82.25% |
| ImageNet | Standard | 75.96% | 63.39% | **31.42%** | 73.24% | 60.42% | 67.90% | 44.98% |
|  | No Crop | 70.81% | 59.09% | **16.52%** | 66.75% | 45.17% | 62.78% | 34.17% |
|  | Aug. 30 | 65.96% | 68.60% | **32.90%** | 70.27% | 45.72% | 69.28% | 47.25% |
|  | Aug. 40 | 66.19% | 67.58% | **33.86%** | 69.50% | 44.60% | 68.88% | 48.72% |
|  | W-10 (30) | 76.14% | 73.19% | **52.76%** | 74.42% | 61.18% | 73.74% | 61.06% |
|  | W-10 (40) | 74.64% | 71.36% | **50.23%** | 72.86% | 59.34% | 71.95% | 59.23% |

**Padding Experiments.** A natural question is whether the reduced accuracy of the models is due to the cropping applied during the transformation. We verify that this is not the case by applying zero and reflection padding to the image datasets. We note that the zero padding creates a "black canvas" version of the dataset, ensuring that no information from the original image is lost after a transformation. We show a random set of adversarial examples in this setting in Figure 6 and a full evaluation in Table 4. We also provide more details regarding reflection padding in Section B and provide an evaluation in Table 6. All of these are in Appendix A.

## 4.2 COMPARING ATTACK METHODS

In Table 2 we compare different attack methods on various classifiers and datasets. We observe that worst-of-10 is a powerful adversary despite its limited interaction with the target classifier. The first-order adversary performs significantly worse. While it is still better than a random transformation , it fails to approximate the ground-truth accuracy of the models and performs significantly worse than the grid adversary and even the worst-of-10 adversary.

**Understanding the Failure of First-Order Methods.** The fact that first-order methods fail to reliably find adversarial rotations and translations is in sharp contrast to previous work on $\ell_p$-robustness (Carlini & Wagner, 2016; Madry et al., 2017). For $\ell_p$-bounded perturbations parametrized directly in pixel space, prior work found the optimization landscape to be well-behaved which allowed first-order methods to consistently find maxima with high loss. In the case of spatial

Table 2: Comparison of attack methods across datasets and models. Worst-of-10 is very effective and significantly reduces the model accuracy despite the limited interaction. The first-order (FO) adversary performs poorly, despite the large number of steps allowed. We compare standard training to Augmentation ($\pm3$px, $\pm30°$). For the full table, see Figure 3 of Appendix A.

|  | MNIST | | CIFAR-10 | | ImageNet | |
|---|---|---|---|---|---|---|
|  | Standard | Aug. | Standard | Aug. | Standard | Aug. |
| Natural | 99.31% | 99.53% | 92.62% | 90.02% | 75.96% | 65.96% |
| Worst-of-10 | 73.32% | 98.33% | 20.13% | 79.92% | 47.83% | 50.62% |
| First-Order | 79.84% | 98.78% | 62.69% | 85.92% | 63.12% | 66.05% |
| Grid | **26.02%** | **95.79%** | **2.80%** | **58.92%** | **31.42%** | **32.90%** |

perturbations, we observe that the non-concavity of the problem is a significant barrier for first-order methods. We investigate this issue by visualizing the loss landscape. For a few random examples from the three datasets, we plot the cross-entropy loss of the examples as a function of translation and rotation. Figure 2 shows one example for each dataset and additional examples are visualized in Figure 11 of the appendix. The plots show that the loss landscape is indeed non-concave and contains many local maxima of low value. The low-dimensional problem structure seems to make non-concavity a crucial obstacle. Even for MNIST, where we observe fewer local maxima, the large flat regions prevent first-order methods from finding transformations of high loss.

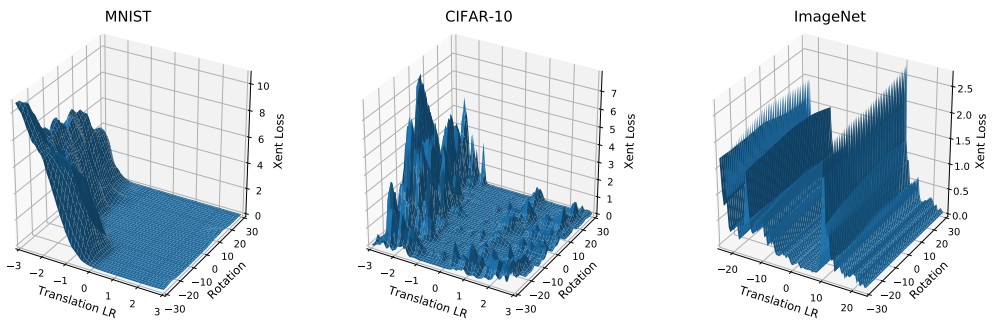

Figure 2: Loss landscape of a random example for each dataset when performing left-right translations and rotations. Translations and rotations are restricted to 10% of the image pixels and $30°$, respectively. We observe that the landscape is significantly non-concave, rendering first-order methods to generate adversarial example ineffective. Figure 11 in the appendix shows additional examples.

**Relation to Black-Box Attacks.** Given its limited interaction with the model, the worst-of-10 adversary achieves a significant reduction in classification accuracy. It performs only 10 *random*, *non-adaptive* queries to the model and is still able to find adversarial examples for a large fraction of the inputs (see Table 2). The low query complexity is an important baseline for black-box attacks on neural networks, which recently gained significant interest (Papernot et al., 2017; Chen et al., 2017; Bhagoji et al., 2017; Ilyas et al., 2017). Black-box attacks rely only function evaluations of the target classifier, without additional information such as gradients. The main challenge is to construct an adversarial example from a small number of queries. Our results show that it is possible to find adversarial rotations and translations for a significant fraction of inputs with very few queries.

**Combining Spatial and $\ell_\infty$-Bounded Perturbations** Table 1 shows that models trained to be robust to $\ell_\infty$ perturbations do not achieve higher robustness to spatial perturbations. This provides evidence that the two families of perturbation are orthogonal to each other. We further investigate this possibility by considering a combined adversary that utilizes $\ell_\infty$ bounded perturbations on top of rotations and translations. The results are shown in Figure 12. We indeed observe that these combined attacks reduce classification accuracy in an (approximately) additive manner.

### 4.3 EVALUATING OUR DEFENSE METHODS.

As we see in Table 1, training with a worst-of-10 adversary significantly increases the spatial robustness of the model, also compared to data augmentation with random transformations. We conjecture that using more reliable methods to compute the worst-case transformations will further improve these results. Unfortunately, increasing the number of random transformations per training example quickly becomes computationally expensive. And as pointed out above, current first-order methods also appear to be insufficient for finding worst-case transformations efficiently.

Our results for majority-based inference are presented in Table 5 of Appendix A. By combining these two defense, we improve the worst-case performance of the models from 26% to 98% on MNIST, from 3% to 82% on CIFAR10, and from 31% to 56% on ImageNet (Top 1).

## 5 RELATED WORK

The fact that small rotations and translation can fool neural networks on MNIST and CIFAR10 was first observed in (Fawzi & Frossard, 2015). They compute the minimum transformation required to fool the model and use it as a measure for a quantitative comparison of different architectures and training procedures. The main difference to our work is that we focus on the optimization aspect of the problem . We show that a few random queries usually suffice for a successful attack, while first-order methods are ineffective. Moreover, we go beyond standard data augmentation and evaluate the effectiveness of natural baseline defenses.

The concurrent work of Kanbak et al. (2017) proposes a different first-order method to evaluate the robustness of classifiers based on geodesic distances on a manifold. This metric is harder to interpret than our parametrized attack space. Moreover, given our findings on the non-concavity of the optimization landscape, it is unclear how close their method is to the ground truth (exhaustive enumeration). While they perform a limited study of defenses (adversarial fine-tuning) using their method, it appears to be less effective than our baseline worst-of-10 training. We attribute this difference to the inherent obstacles first-order methods face in this optimization landscape.

Recently, Xiao et al. (2018) and Tramèr & Boneh (2017) observed independently that it is possible to use various spatial transformations to construct adversarial examples for naturally and adversarially trained models. The main difference from our work is that we show even very simple transformations (translations and rotations) are sufficient to break a variety of classifiers, while the transformations employed in (Xiao et al., 2018) and (Tramèr & Boneh, 2017) are more involved. The transformation in (Xiao et al., 2018) is based on performing a displacement of individual pixels in the original image constrained to be globally smooth and then optimized for misclassification probability. Tramèr & Boneh (2017) consider an $\ell_\infty$-bounded pixel-wise perturbation of a version of the original image that has been slightly rotated and in which a few random pixels have been flipped. Both of these methods require direct access to the attacked model (or a surrogate) to compute (or at least estimate) the gradient of the loss function with respect to the model's input. In contrast, our attacks can be implemented using only a small number of random, non-adaptive transformations of the input.

## 6 CONCLUSIONS

We examined the robustness of state-of-the-art image classifiers to translations and rotations. We observed that even a small number of randomly chosen perturbations of the input are sufficient to considerably degrade the classifier's performance.

The fact that common neural networks are vulnerable to simple and naturally occurring spatial transformations (and that these transformations can be found easily from just a few random tries) indicates that adversarial robustness should be a concern not only in a fully worst-case security setting. We conjecture that additional techniques need to be incorporated in the architecture and training procedures of modern classifiers to achieve worst-case spatial robustness. Also, our results underline the need to consider broader notions of similarity than only pixel-wise distances when studying adversarial misclassification attacks. In particular, we view combining the pixel-wise distances with rotations and translations as a next step towards the "right" notion of similarity in the context of images.

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

# A    OMITTED TABLES AND FIGURES

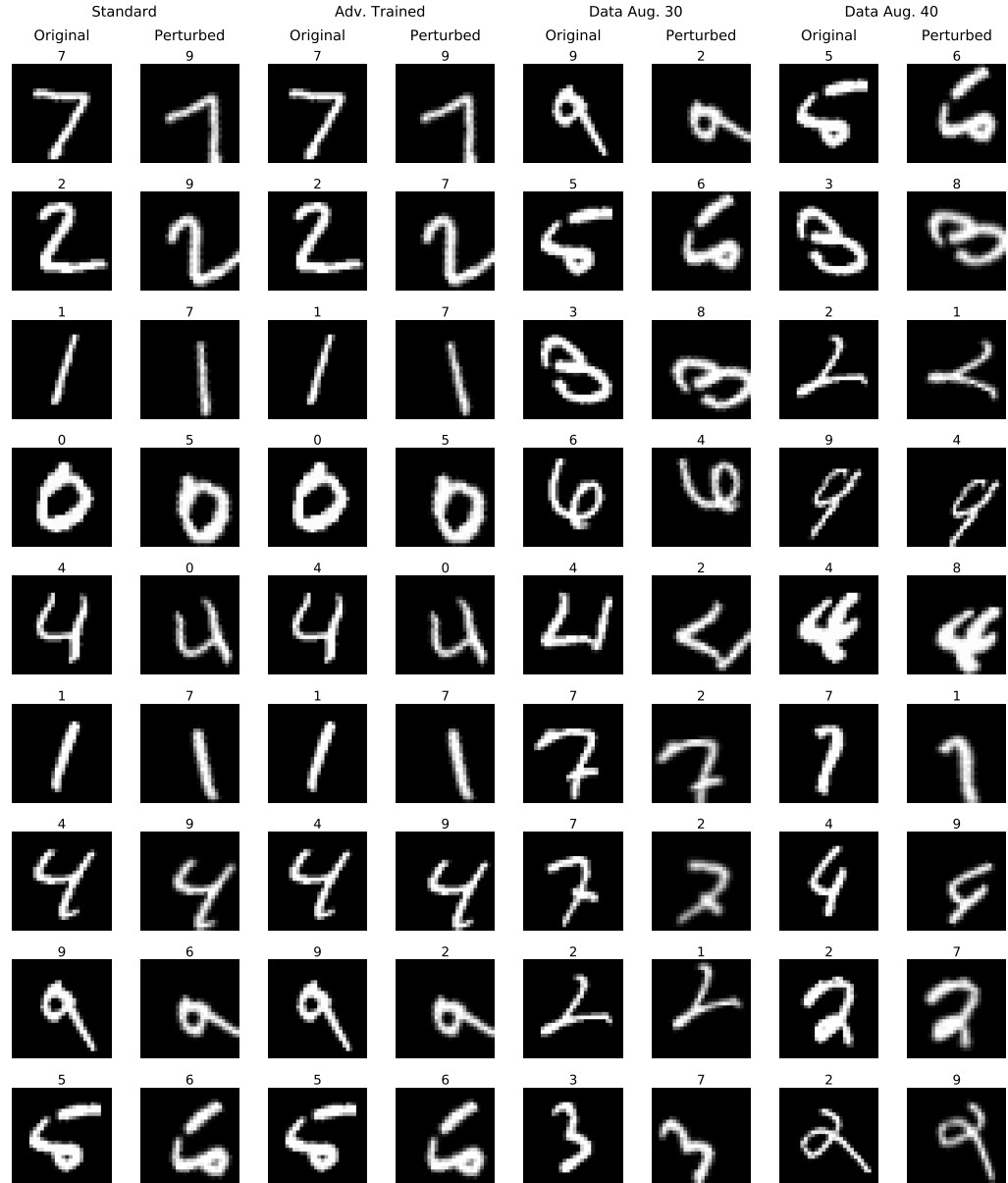

Figure 3: MNIST. Successful adversarial examples for the models studied in Section 4. Rotations are restricted to be within 30° of the original image and translations up to 3 pixels per direction (image size 28 × 28). Each example is visualized along with its predicted label in the original and perturbed versions.

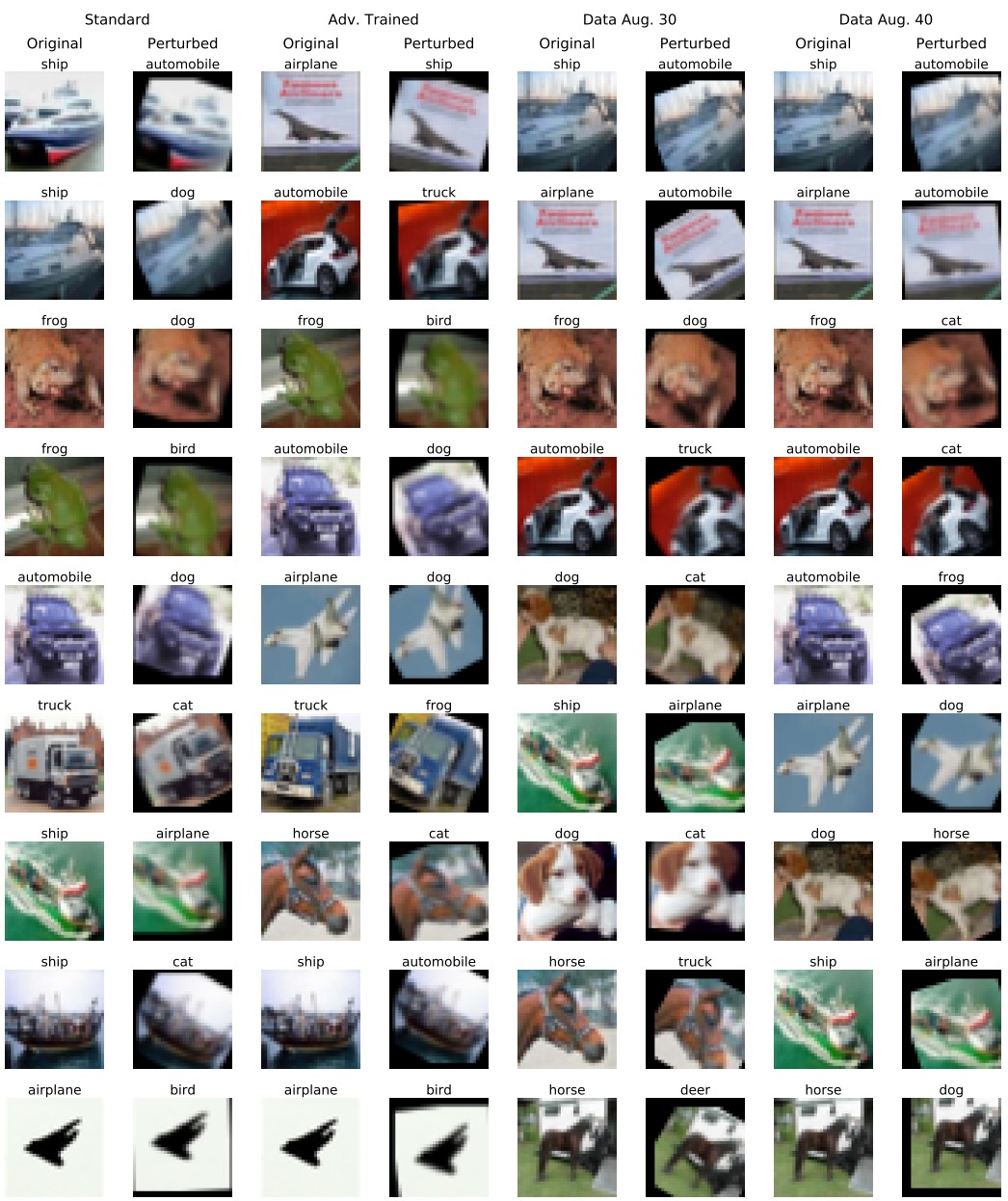

Figure 4: CIFAR10. Successful adversarial examples for the models studied in Section 4. Rotations are restricted to be within $30°$ of the original and translations up to 3 pixels per directions (image size $32 \times 32$). Each example is visualized along with its predicted label in the original and perturbed version.

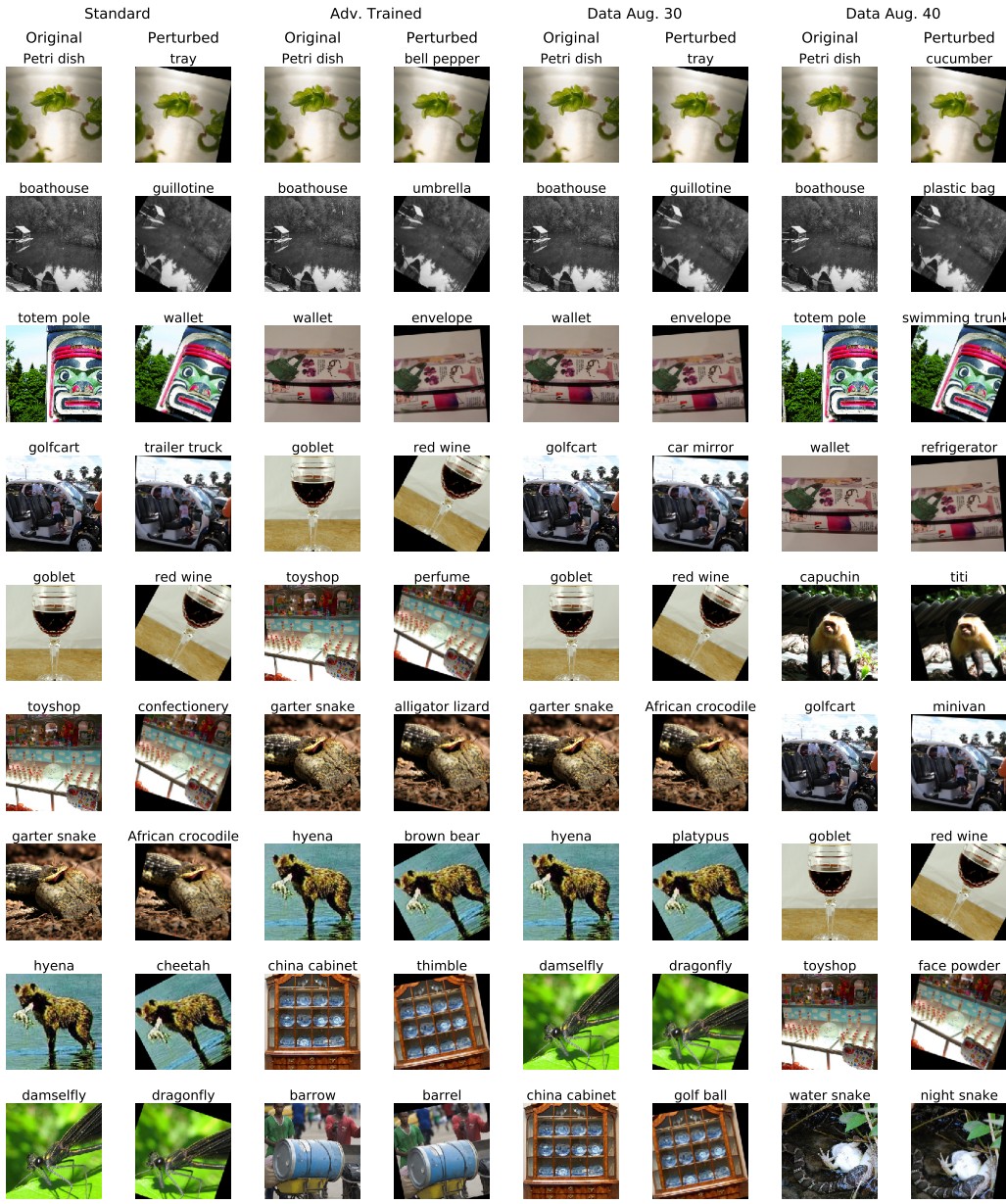

Figure 5: ImageNet. Successful adversarial examples for the models studied in Section 4. Rotations are restricted to be within $30°$ of the original and translations up to $24$ pixels per directions (image size $299 \times 299$). Each example is visualized along with its predicted label in the original and perturbed version.

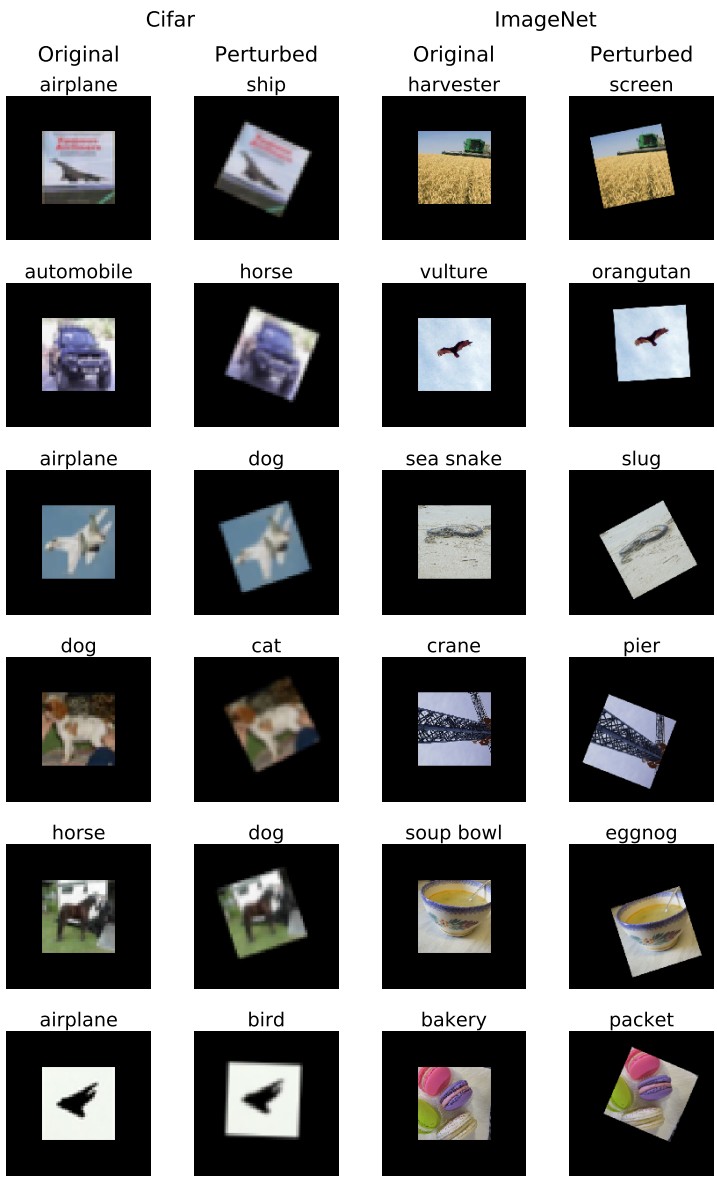

Figure 6: Sample adversarial transformations for the "black-canvas" setting for the standard models on CIFAR10 and ImageNet.

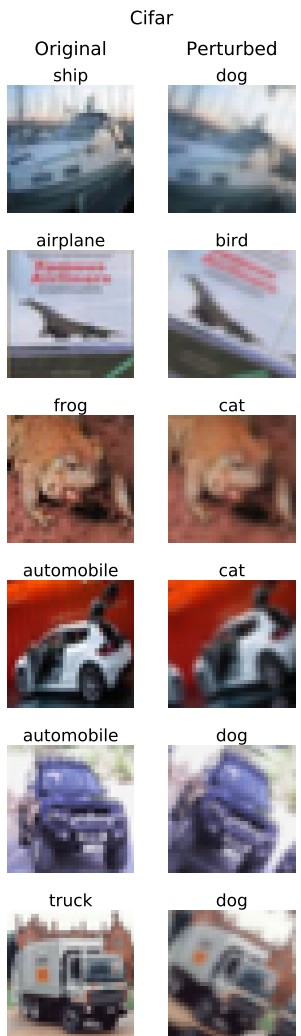

Figure 7: Sample adversarial transformations for the reflection padding setting for the standard models on CIFAR10.

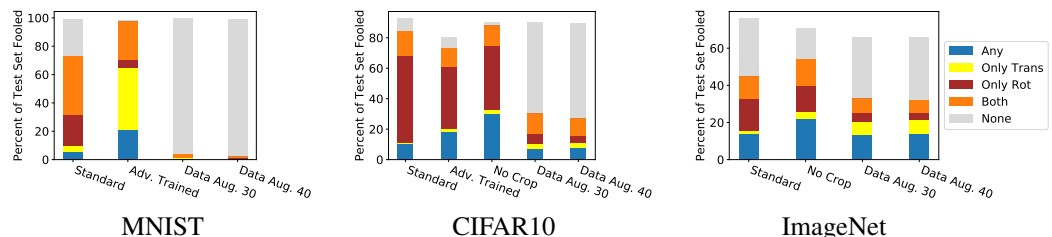

Figure 8: Fine-grained dataset analysis. For each model, we visualize what percent of the test set can be fooled via various methods. We compute how many examples can be fooled with either translations or rotations ("any"), how many can be fooled only by one of these, and how many require a combination to be fooled ("both").

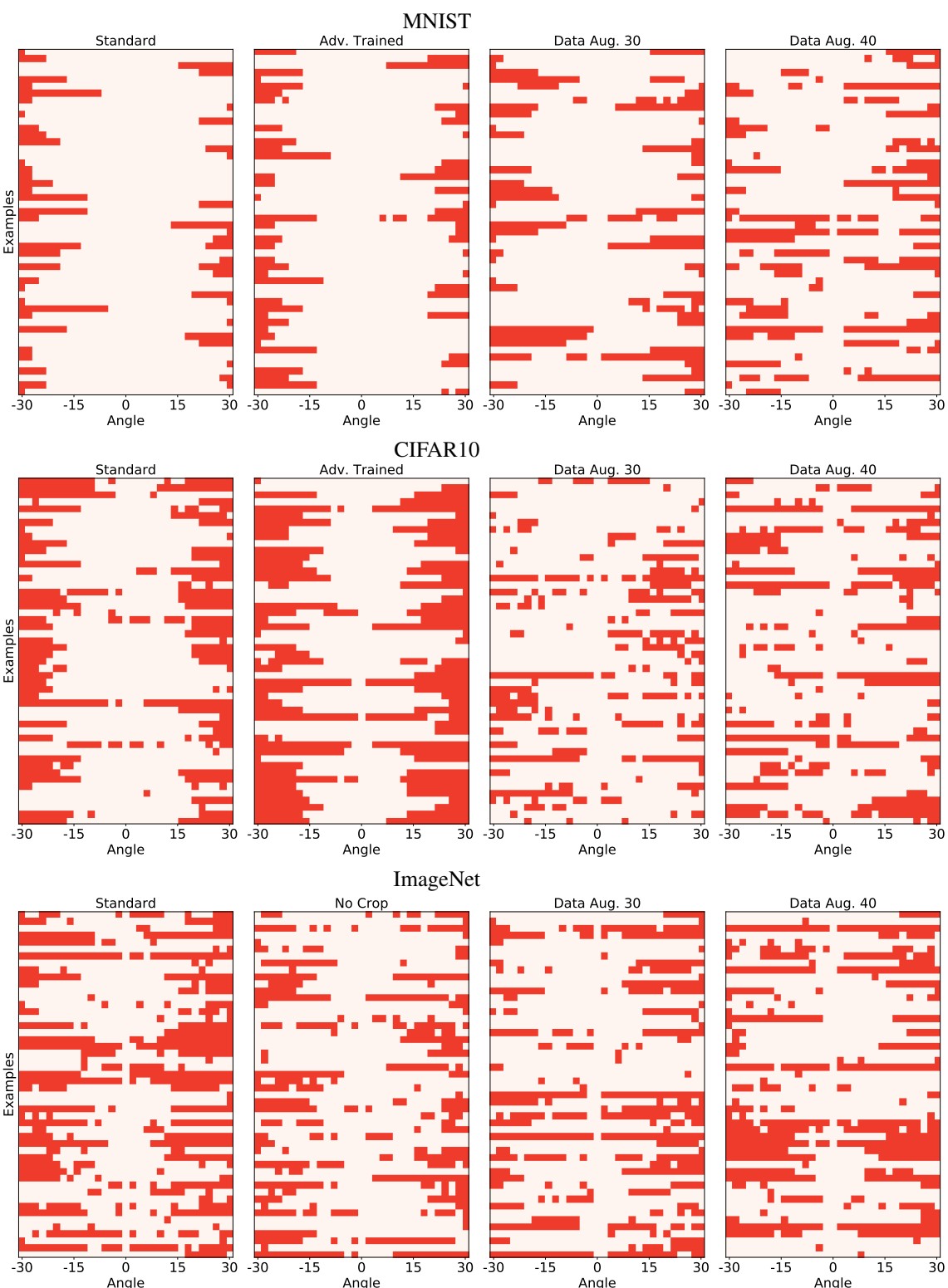

Figure 9: Visualizing which angles fool the classifier for 50 random examples. For each dataset and model, we visualize one example per row. Red corresponds to *misclassification* of the images. We observe that the angles fooling the models form a highly non-convex set.

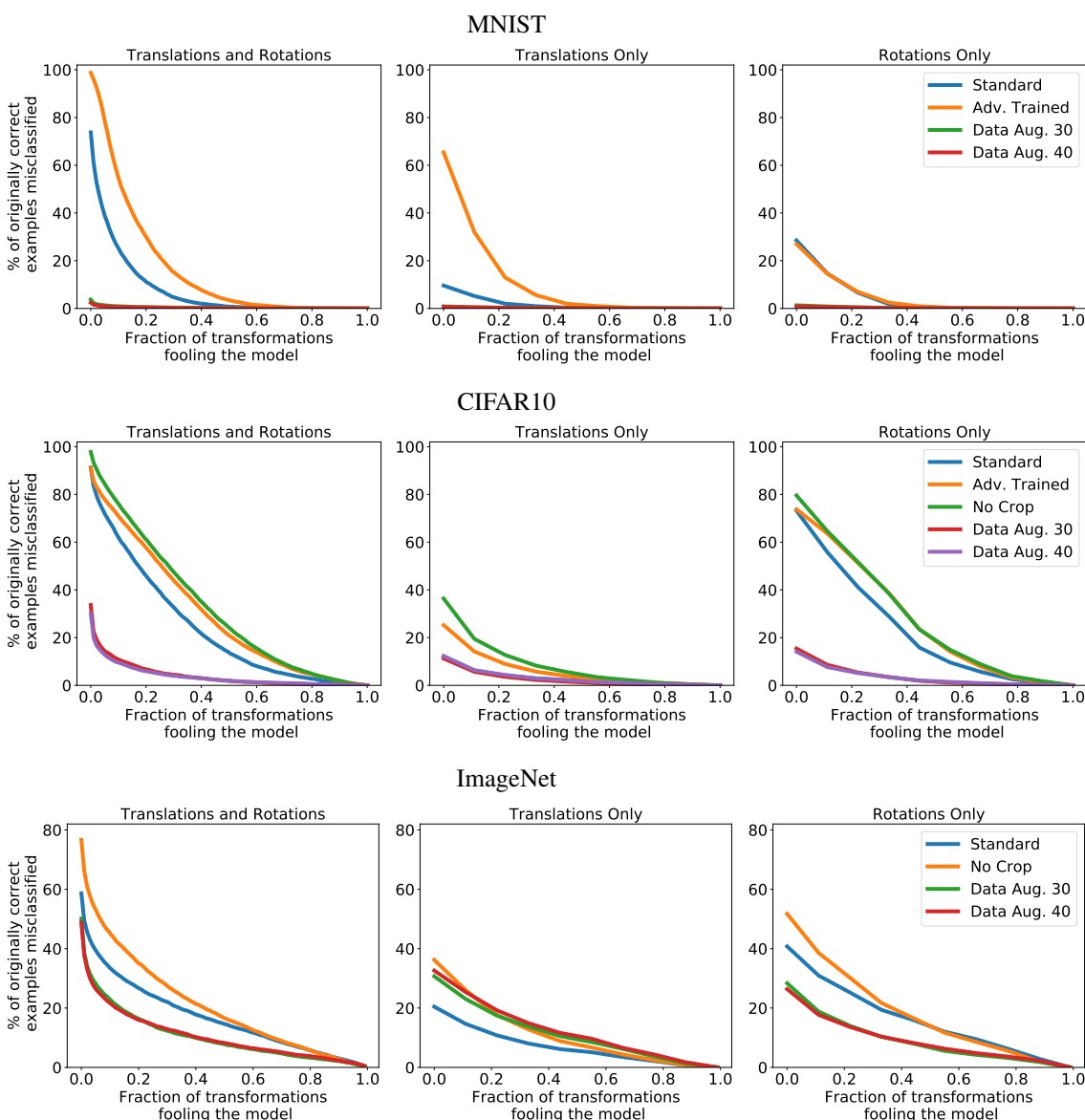

Figure 10: Cumulative Density Function plots. For each fraction of grid points $p$, we plot the percentage of correctly classified test set examples that are fooled by at least $p$ of the grid points. For instance, we can see from the first plot, MNIST Translations and Rotations, that approximately 10% of the correctly classified natural examples are misclassified under $1/5$ of the grid points transformations.

Table 3: Comparison of attack methods across datasets and models.

|  | Model | Natural | Worst-of-10 | FO | Grid |
|---|---|---|---|---|---|
| **MNIST** | Standard | 99.31% | 73.32% | 79.84% | **26.02%** |
|  | $\ell_\infty$-Adversarially Trained | 98.65% | 51.18% | 81.23% | **1.20%** |
|  | Aug. 30 ($\pm$3px, $\pm$30°) | 99.53% | 98.33% | 98.78% | **95.79%** |
|  | Aug. 40 ($\pm$4$\pm$, $\pm$40°) | 99.34% | 98.49% | 98.74% | **96.95%** |
| **CIFAR10** | Standard | 92.62% | 20.13% | 62.69% | **2.80%** |
|  | No Crop | 90.34% | 15.04% | 52.27% | **1.86%** |
|  | $\ell_\infty$-Adversarially Trained | 80.21% | 19.38% | 33.24% | **6.02%** |
|  | Aug. 30 ($\pm$3px, $\pm$30°) | 90.02% | 79.92% | 85.92% | **58.92%** |
|  | Aug. 40 ($\pm$4px, $\pm$40°) | 88.83% | 80.47% | 85.48% | **61.69%** |
| **ImageNet** | Standard | 75.96% | 47.83% | 63.12% | **31.42%** |
|  | No Crop | 70.81% | 35.52% | 55.93% | **16.52%** |
|  | Aug. 30 ($\pm$24px, $\pm$30°) | 65.96% | 50.62% | 66.05% | **32.90%** |
|  | Aug. 40 ($\pm$32px, $\pm$40°) | 66.19% | 51.11% | 66.14% | **33.86%** |

Table 4: Evaluation of a subset of Table 1 in the "black-canvas" setting (images are zero-padded to avoid cropping due to rotations and translations). The models are trained on padded images.

|  |  | Natural | Random | Worst-of-10 | Grid | Trans. Grid | Rot. Grid |
|---|---|---|---|---|---|---|---|
| **CIFAR10** | Standard | 91.81% | 70.23% | 25.51% | **6.55%** | 83.38% | 12.44% |
|  | No Crop | 89.70% | 52.86% | 14.14% | **1.17%** | 47.94% | 9.46% |
|  | Aug. 30 ($\pm$3px, $\pm$30°) | 91.45% | 90.82% | 80.53% | **63.64%** | 82.28% | 76.32% |
|  | Aug. 40 ($\pm$4px, $\pm$40°) | 91.24% | 91.00% | 81.81% | **66.64%** | 81.75% | 78.57% |
| **ImageNet** | Standard | 73.60% | 46.59% | 29.51% | **15.38%** | 28.03% | 23.81% |
|  | No Crop | 66.28% | 38.70% | 14.17% | **3.43%** | 8.87% | 10.97% |
|  | Aug. 30 ($\pm$24px, $\pm$30°) | 64.60% | 67.75% | 47.32% | **28.51%** | 45.33% | 39.33% |
|  | Aug. 40 ($\pm$32px, $\pm$40°) | 49.20% | 57.69% | 38.36% | **22.10%** | 32.84% | 32.95% |

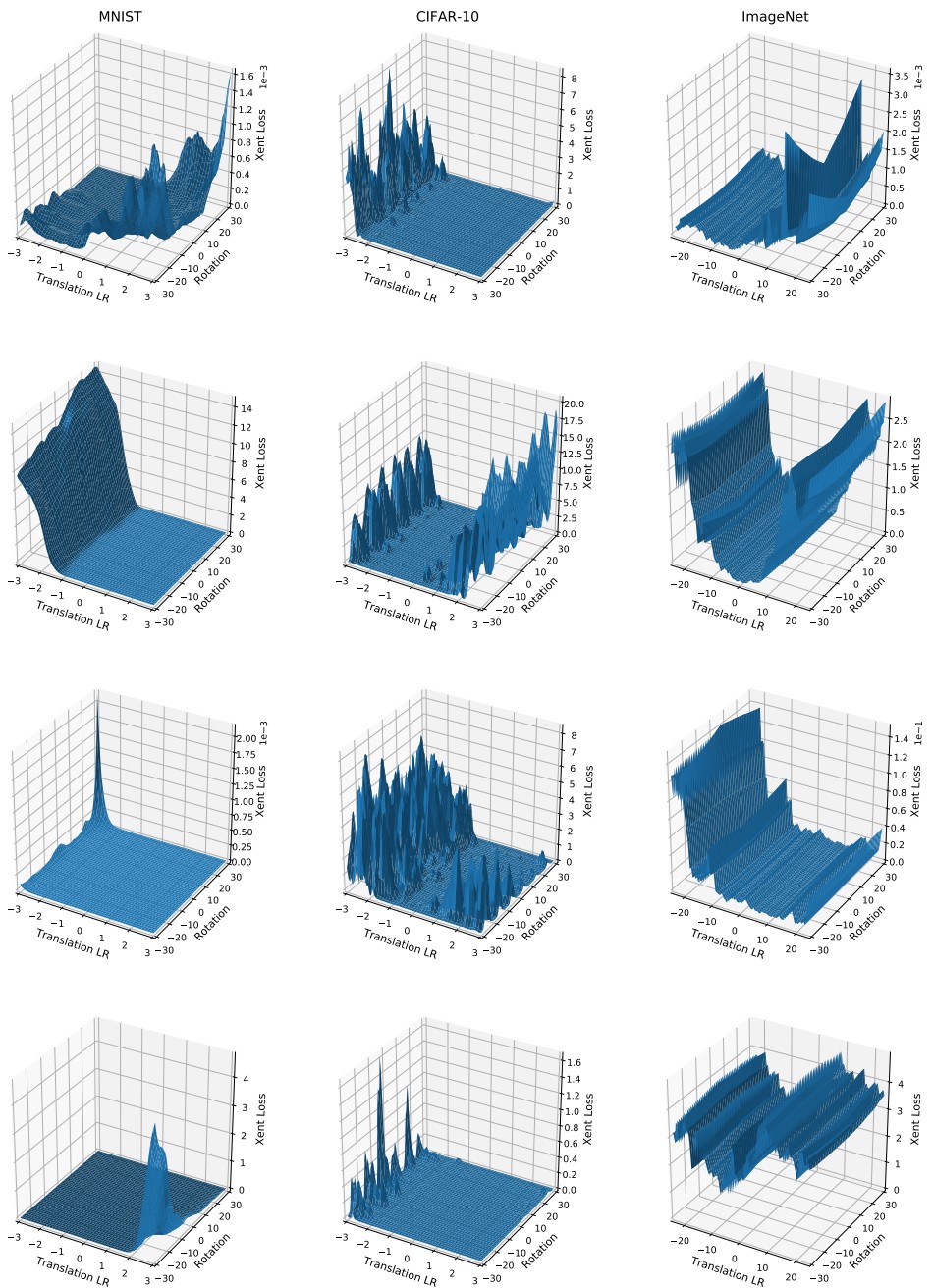

Figure 11: Loss landscape of 4 random examples for each dataset when performing left-right translations and rotations. Translations and rotations are restricted to 10% of the image pixels and 30° respectively. We observe that the landscape is significantly non-concave, making rendering FO methods for adversarial example generation powerless.

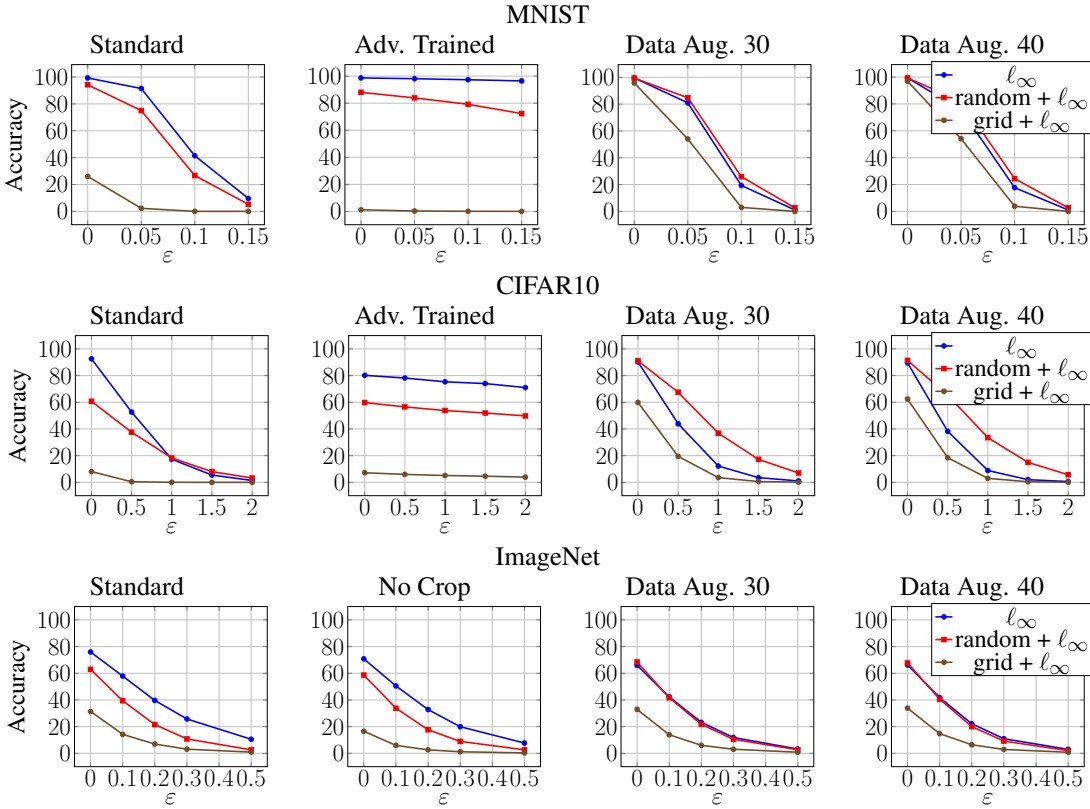

Figure 12: Accuracy of different classifiers against $\ell_\infty$-bounded adversaries with various values of $\varepsilon$ and spatial transformations. For each value of $\varepsilon$, we perform PGD to find the most adversarial $\ell_\infty$-bounded perturbation. Additionally, we combine PGD with random rotations and translations and with a grid search over rotations and translations in order to find the transformation that combines with PGD in the most adversarial way.

## B    MIRROR PADDING

In the experiments of Section 4, we filled the remaining pixels of rotated and translated images with black (also known as zero or constant padding). This is the standard approach used when performing random cropping for data augmentation purposes. We briefly examined the effect of mirror padding, that is replacing empty pixels by reflecting the image around the border[6]. The results are shown in Table 6. We observed that training with one padding method and evaluating using the other resulted in a significant drop in accuracy. Training using one of these methods randomly for each example resulted in a model which roughly matched the best-case of the two individual cases.

---

[6]https://www.tensorflow.org/api_docs/python/tf/pad

Table 5: Majority Defense. Accuracy of different models on the natural evaluation set and against a combined rotation and translation adversary using aggregation of multiple random transformations.

|  |  | Natural Acc. | | Grid Acc. | |
| --- | --- | --- | --- | --- | --- |
|  | Model | Stand. | Vote | Stand. | Vote |
| MNIST | Standard | 99.31% | 98.71% | 26.02% | 18.80% |
|  | Aug 30. | **99.53**% | 99.41% | 95.79% | 95.32% |
|  | Aug 40. | 99.34% | 99.25% | 96.95% | 97.65% |
|  | W-10 (30) | 99.48% | 99.40% | 97.32% | 96.95% |
|  | W-10 (40) | 99.42% | 99.41% | 97.88% | **98.47**% |
| CIFAR10 | Standard | 92.62% | 80.37% | 2.82% | 7.85% |
|  | Aug 30. | 90.02% | 92.70% | 58.90% | 69.65% |
|  | Aug 40. | 88.83% | 92.50% | 61.69% | 76.54% |
|  | W-10 (30) | 91.34% | 93.38% | 69.17% | 77.33% |
|  | W-10 (40) | 91.00% | **93.40**% | 71.15% | **81.52**% |
| ImageNet | Standard | 75.96% | 73.19% | 31.42% | 40.21% |
|  | Aug 30. | 65.96% | 72.44% | 32.90% | 44.46% |
|  | Aug 40. | 66.19% | 71.46% | 33.86% | 46.98% |
|  | W-10 (30) | **76.14**% | 74.92% | 52.76% | **56.45**% |
|  | W-10 (40) | 74.64% | 73.38% | 50.23% | 56.23% |

|  | Natural | Random (Zero) | Random (Mirror) | Grid Search (Zero) | Grid Search (Mirror) |
| --- | --- | --- | --- | --- | --- |
| Standard Nat | 92.62% | 60.76% | 66.42% | 8.08% | 5.37% |
| Standard Adv | 80.21% | 59.79% | 67.12% | 7.20% | 12.89% |
| Aug. A, Zero | 90.25% | 91.09% | 87.67% | 59.87% | 40.55% |
| Aug. B, Zero | 89.55% | 91.40% | 87.94% | 62.42% | 42.37% |
| Aug. A, Mirror | 92.25% | 88.43% | 91.05% | 41.46% | 53.95% |
| Aug. B, Mirror | 92.03% | 88.58% | 91.34% | 45.44% | 57.97% |
| Aug. A, Both | 91.80% | 90.98% | 91.28% | 56.95% | 52.60% |
| Aug. B, Both | 91.57% | 91.87% | 91.11% | 60.46% | 56.13% |

Table 6: CIFAR10: The effect of using reflection or zero padding when training a model. The experimental setup matches that of Section 4. Zero padding refers to filling the empty pixels caused by translations and rotations with black. Mirror padding corresponds to using a reflection of the images. "Both" refers to training using both methods and alternating randomly between them for each training example.

