# OpenReview forum: "A Rotation and a Translation Suffice: Fooling CNNs with Simple Transformations"
_ICLR.cc/2019/Conference_

### Official Review · AnonReviewer3 · 2018-10-26
**Interesting approach but not yet matured enough.**

**Rating:** 5
**Confidence:** 4

**Review:**

Summary
The authors study robustness of neural networks for image recognition tasks with respect to geometric transformations in input space. The question is posed in an adversarial setting, where the authors exploit that Conv/ResNets are not fully translation and rotation invariant. The authors propose three untargeted attacks to increase the classification error of the network: a first-order method, an attack involving random transformations and a grid search of allowed transformations. For the random and grid search the worst prediction is considered the outcome of the attack. The authors observe that first-order attacks are not very successful in fooling the network compared to the grid search. Data augmentation as a counter measure is found to be not sufficient and adversarial (robust) training with respect to the random search attack is proposed in addition.

Evaluation
The paper is well written and particularly the empirical part is interesting. However, novelty is limited, the best approach boils down to a grid search that tests multiple hypotheses instead of a single one. There are some conceptual problems and important aspects like confidences of the classification are not addressed.

Novelty:
Many claims and observations appear trivial and well-known. E.g.,
- the research question has already been addressed by related work, leaving the proposed attacks trivial given that the attack space (allowed transformations) is specified ad hoc and without a proper measure.
- that data augmentation and training with the adversarial loss function (i.e. with the attack scheme in mind) is helpful is straight forward and not surprising

Detailed comments:
The authors study whether neural networks are robust to transformations in input space and resort to a benign adversarial setting. I'm wondering whether this allows for an answer regarding general robustness? That is, the experiments are conducted wrt the worst case, while the training does not account for an attack setting. E.g, it is unclear why the classifier would not involve a pre-processing step to counter transformations in input space, see Rowley et al. (1998).

Translation and rotation invariance of neural networks has been addressed by many authors, e.g., see Jaderberg et al. (2015) and Marcos et al. (2017).

Adversarial examples are defined to be similar and misclassified with high confidence.
The similarity of the transformation is not addressed properly. E.g., if the goal of the adversary is to force errors, why not allow for rotations of 180 degrees? Pixel-based attacks (Goodfellow et al., 2014) are more rigorous in this regard while the cited transformation-based attacks (Kanbak et al. 2017; Xiao et al., 2018) are virtually indistinguishable from the real test cases.

The effectiveness of the grid search attack seems to be connected to performing $5 * 5 * 31 = 775$ individual tests for each test case where only the worst outcome would count. The sheer number should render a misclassification more likely compared to the competitors. This is supported by empirical findings showing that only a small subset of transformations per test case accounts for the misclassification on CIFAR10 and ImageNet (Fig. 10 in the Appendix).

Regarding the padding experiments, I wonder whether the network architecture is appropriate for the new input. Here, more experimentation is necessary. The conclusion with respect to the first-order method remains a conjecture.

References:
- Ian J. Goodfellow, Jonathon Shlens, and Christian Szegedy. Explaining and harnessing adversarial examples. arXiv preprint arXiv:1412.6572, 2014.
- Max Jaderberg, Karen Simonyan, and Andrew Zisserman. Spatial transformer networks. In Advances in neural information processing systems, pp. 2017–2025, 2015.
- Can Kanbak, Seyed-Mohsen Moosavi-Dezfooli, and Pascal Frossard. Geometric robustness of deep networks: analysis and improvement. arXiv preprint arXiv:1711.09115, 2017.
- Diego Marcos, Michele Volpi, Nikos Komodakis, and Devis Tuia. Rotation equivariant vector field networks. In The IEEE International Conference on Computer Vision (ICCV), Oct 2017.
- Henry Rowley, Shumeet Baluja, and Takeo Kanade. Rotation invariant neural network-based face detection. In Proceedings of IEEE Conference on Computer Vision and Pattern Recognition, pp. 38. sn, 1998.
- Chaowei Xiao, Jun-Yan Zhu, Bo Li, Warren He, Mingyan Liu, and Dawn Song. Spatially transformed adversarial examples. arXiv preprint arXiv:1801.02612, 2018.

---

> ### Author Response · Authors · 2018-11-26
> **Author response (part 1)**
>
> We thank the reviewer for their comments. We will address concerns raised below:
>
> - On the novelty of our approach. We are aware that the robustness of NNs to spatial transformations has already been addressed as a research question. We cite some of the relevant work (collecting all the citations in a vast field such as deep learning is unfortunately hardly possible) and we plan to add the references pointed out by the reviewer. However, we believe that our paper contains a number of novel contributions that are not present in prior work (as we outline in our "summary of contributions" section):
> a) We evaluate concrete and natural baselines for improving spatial robustness. While our worst-of-10 training approach is inspired by robust optimization, it has not been applied in this form before (to the best of our knowledge).
> b) We evaluate the robustness of L_inf trained models, as well as combinations of L_inf and spatial attacks.
> c) We provide insights on the performance of first-order methods for this task.
> d) Finally, we point out that a few black-box queries suffice to find misclassifications.
> (Please also refer to our response to Reviewer 2 on the same topic.)
>
> - The reviewer mentions multiple times how our results are trivial and not surprising. We believe that when a problem is as fundamental as the one we study, rigorous evaluation of simple baselines is an important contribution. We were not able to find such a thorough study in prior work.
>
> - We are aware of the existence of rotation-invariant architectures as well as preprocessing approaches to transformation robustness. Our goal is to understand the spatial robustness of the most popular architectures used for these tasks and challenge the conventional wisdom that these architectures will automatically generalize to naturally occuring transformations. Regarding the results of Rowley et al. (1998), we would like to emphasize that rotation-invariant face detection is significantly easier, since faces have a canonical orientations (whereas most CIFAR10 and ImageNet categories do not). The goal of Marcos et al. (2017) is to learn models that are *fully* rotation invariant (i.e. 180 degree rotations are acceptable). This is a fundamentally different task from learning models robust to *small* rotations since the later can still exploit the orientation of the image to some extent. We will add a discussion of these papers in our manuscript.
>
> - The definition of adversarial examples we are concerned with is "images with semantically similar content that are classified differently". Arguably, a 30 degree rotation does not change the content of the image to a human (see provided images in the Appendix), so we consider these to be valid adversarial examples. In contrast, a 180 degree rotation would certainly appear wrong or unnatural to a human, such as a '6' turned into a '9' or an upside-down boat. We don't view the "high confidence" component of "standard" adversarial examples as very important. After all, classifiers are evaluated based on their top1 (and sometimes top5) accuracy. Thus the relevant quantity in our setting is the robust accuracy. Similarly, we do not view indistinguishability as important, as long as the original and perturbed images have the same content to a human and do not appear clearly transformed.
>
> - On related work: We want to emphasize that the attack of Xiao et al. (2018) is a complex attack, operating in a much larger perturbation space (the space of smooth deformations) and utilizing *full gradient access* to the target model. It is natural to expect such an attack to succeed with high confidence with minimal distortion. At the same time, however, these cannot be seen as "naturally occuring transformations" outside of a security context, in contrast to rotations and translations.
>
> We do not follow the reviewers’ comments about the transformations of Kanbak et al. (2018). The set of transformations from their paper is very similar to ours, differing in only in parametrization. We also want to emphasize that their paper focuses exclusively on first order methods. Thus the reliability of their evaluation (compared to the ground-truth gridding that we perform) is questionable given our results about the effectiveness of first-order methods. We already include a discussion about this paper in our manuscript.
>
> - We do not understand the reviewer's concern about the amount of grid points used and would be grateful if the reviewer could further elaborate this point. If the loss landscape was as well-behaved as in the case of L_inf perturbations, an attack based on projected gradient descent would clearly match a grid search. After all, PGD is allowed full access to the gradient of the model (3 real numbers) whereas our grid attack only has access to the output label of the model (a single bit, "classified correctly" or "misclassified").

---

> > ### Author Response · Authors · 2018-11-26
> > **Author response (part 2)**
> >
> > - Regarding our padding experiments: These experiments act as a simple sanity check to make sure that the accuracy lost is not due to losing information from the edges of the image. The accuracy of the models in this setting with standard training is very close to that of the original dataset (Table 4) so we believe the our architecture is at least adequate for these tasks.
> >
> > - We are concerned about the reviewers skepticism regarding our first-order results without providing concrete arguments. We provide empirical evidence showing that first-order attacks are not sufficient to reduce the accuracy of these models. We additionally visualize the optimization landscape for multiple inputs from multiple datasets and we observe significant non-concavity. Does the reviewer have concrete suggestions about what additional evidence would be convincing?

---

### Official Review · AnonReviewer2 · 2018-11-05
**some interesting observations**

**Rating:** 6
**Confidence:** 2

**Review:**

The paper states that basic transformation (translation and rotation) can easily fool a neural network in image classification tasks. Thus, image classification models are actually more vulnerable than people thought. The message conveyed by the paper is clear and easy to get. The experiments are natural and interesting. Some interesting points:
  --The model trained with data augmentation that covers the attack space does not alleviate the problem sufficiently.
  --Gradient descent does not provide strong attack, but grid search does. This may be due to the high non-concavity, compared to the small perturbation case.

One possible question is the novelty, as this idea is so simple that probably many people have observed similar phenomenon--but have not experimented that extensively.
Also, there are some related works that also show the vulnerability under spatial transformations. But some are concurrent works to 1st version of the paper (though published), so I tend to not to judge it by those works.

Other comments:
1. page 3 in the paragraph starting with ‘We implement …’, the author chooses a differentiable bilinear interpolation routine. However, the interpolation method is not shown or explained.
2. In term of transformation, scaling and reflecting are also transformations. It should be straightforward to check the robustness with respect to them. Comments?
3. Header in tables is vague. Like ‘Natural’ or ‘Original’, etc. More description of the Header under tables is helpful.
4. For CIFAR10 and especially for ImageNet dataset, Aug30 and Aug40 models showed lower accuracy than No Crop model on Nat test set. This is little strange because data augmentation (such as random rotation) is commonly used strategy to improve test accuracy. I think this might mean that the model is not trained enough and underfitted, maybe because excessive data augmentation lowered the training speed.

---

> ### Author Response · Authors · 2018-11-26
> **Author response**
>
> We thank the reviewer for their kind words.
>
> Regarding the novelty of our paper: We agree that we are not the first to experimentally study the robustness of classifiers to rotations and translation (as we mention in our paper including relevant citations). We would like to emphasize however that simply pointing out this flaw is not enough to establish it as a relevant problem with current classifiers. After all, we also need to understand if this issue is only a small glitch that we can fix with a few simple modifications, or if it requires more thought and further research. This is why we go into significantly more depth than prior work with respect to possible fixes and show that standard approaches (data augmentation, robust optimization, ensembles / majority voting) do help to some extent, but are still far from fully solving the problem. (Please also refer to our response to Reviewer 3 on the same topic.)
>
> We will address the other points raised below.
>
> 1. We used the approach from "Spatial Transformer Networks" (Jaderberg et al., 2015) and their open source implementation. We will clarify this point in our updated manuscript and add a link to the implementation.
>
> 2. We agree that scaling and reflecting are natural transformations to consider. We decided to restrict ourselves to two transformations and perform a comprehensive study in this case, rather than perform fewer experiments with more transformations. We chose translations since ConvNets are often claimed to be inherently robust to these transformations, and rotations since we believe they are the simplest to describe. Moreover, rotations don't discard any image information (other than edge effects) while (say) downscaling does.
>
> 3.  We will update the manuscript to clarify the table headers.
>
> 4. To the best of our knowledge, none of the publicly available implementations for training state-of-the-art ImageNet or CIFAR10 models use random rotations as data augmentation. This is likely due to the fact that random rotations typically do not yield  any benefits in (non-robust) test error (as our experimental results show).
>
> We would also like to emphasize that all of our models were trained until convergence (the loss plateaued) and hence the reduced test performance is not an artifact of training for insufficient steps (we will add a note about this to our manuscript). At a high level, a decrease in test performance due to data augmentation is not surprising. If the transformations used are not present in the test set, then a model that has learned to be invariant to these transformations will typically perform worse on the test set. (This is also the case when learning models that are adversarially robust to L_p perturbations; there is a decrease in test accuracy.) As an extreme, consider an MNIST model that learns to be invariant to rotations up to 180 degrees. Clearly, this invariance is only hurting the model's performance since it cannot easily distinguish '6' from '9'.

---

### Official Review · AnonReviewer1 · 2018-11-05
**Solid experimental study of approximate worst and average case input image rotation and translation**

**Rating:** 8
**Confidence:** 3

**Review:**

Summary:
Standard CNN models for MNIST, CIFAR10 and ImageNet are vulnerable with regard
to (adversarial) rotation and translation of images.
The paper experimentally examines different ways of formulating attacks
(gradient descent, grid search and sampling) and defenses
(random augmentation, worst-case out of sample robust training,
aggregated classification) for this class of image transformation.

The main results are:
- Gradient descent is not effective at generating worst-case rotations /
translations due to nonconcavity of the adversarial objective
- Grid search is very effective due to low parameter space
- Sampling and pick the worst is also effective and cheap, for similar reasons
- L infinity ball pixel perturbation robustness is orthogonal to the examined
transformations and does not provide good defense mechanism
- Just augmenting data with random translation / rotations is not a strong
defense
- Using a worst-case out of sample of 10 for training with an approximation of
a robust optimization objective combined with an aggregated result for
classification is a stronger defense

Recommendation:
The paper presents a comprehensive study of a relevant class of adversarial
image perturbations for state-of-the-art neural network models.
The results are a useful pointer towards future research directions and for
building more robust systems in practice.
I recommend to accept the paper.

Strong points:
- The paper is well written, has clear structure and is technically easy to
understand.
- The question of padding and cropping comes up naturally and is then answered.

Open questions (things that could potentially be of interest when added):
- Loss landscapes look like most of the nonconcavity is along the translation
parameter. Any idea why?
- What mechanisms within CNN models do or do not learn (generalize) rotation
and translation from provided data (including augmentation)?

Specific:
- Page 2: perturbrbations (Typo)
- Page 3: witho (Typo)
- Page 3: Constrained optimization problems typically written as
max_{...} \mathcal L(x', y) s.t. x' = T(...)
(s.t. for subject to instead of for) but that's matter of taste I guess
- Page 4: first order -> first-order (consistency)
- Page 4: tyipcally (Typo)
- Page 4: occurs most common(ly)

I am not sufficiently knowledgable about the previous literature to ensure that
the claimed novelty of the paper is truly as novel.

---

> ### Author Response · Authors · 2018-11-26
> **Author response**
>
> We thank the reviewer for their kind words.
>
> -- Regarding the loss landscape: it appears that the loss landscape exhibits significant non-concavity in both directions (see page 19 for additional plots). We do agree however that the translation direction is mostly responsible for the loss value (changes along the rotation direction appear jagged and without consistent patterns). We do not have concrete ideas about why this this happening but we agree that it is an interesting research direction.
>
> -- Since for vanilla convolutional layers input translation can only lead to output translation, the main mechanisms responsible for translation non-robustness are max-pooling units and strided convolutions. We do not have a similar understanding for the case of rotations. This is an important direction for future work, but we believe that it goes beyond the scope of the current paper.
>
> We will update our manuscript to incorporate the typos and fixes pointed out.

---

### Public Comment · (anonymous) · 2018-11-27
**Missed related work on fooling CNNs with simple geometric perturbations**

The paper is missing what seems to be a highly related piece of work:

Towards Practical Verification of Machine Learning: The Case of Computer Vision Systems
Kexin Pei, Yinzhi Cao, Junfeng Yang, Suman Jana
https://arxiv.org/abs/1712.01785

The above paper also shows how to use simple geometric transformations such as rotations, translation and others into fooling CNNs.

It would be great if the authors explain what the differences are?

---

> ### Author Response · Authors · 2018-12-04
> **Author response**
>
> We thank the anonymous commenter for bringing this work to our attention. There are several differences between this and our work. We believe that their measure of robustness is less intuitive than ours, but this is a subjective point.
>
> In terms of methodological differences, the way that the authors calculate rotations and translations is less natural than ours; they do not perform interpolation and instead adopt a nearest neighbors approach (from the images in the paper, it is possible to see the jagged edges induced by this). In addition, the authors consider only a very small range of transformations, and do not attempt to uncover the distribution of fooling transformations (and nor do they consider rotations and translations together).
>
> Once again we want to emphasize that our work goes significantly beyond identifying the model vulnerability, but thoroughly evaluates natural methods for improving it. Moreover we want to mention that the suggested paper appeared online within two days of the day that the original version of our work appeared online.

---

### Meta-Review · Area_Chair1 · 2018-12-13
**Interesting observations. Novelty limited. Related work missing.**

**Confidence:** 4
**Recommendation:** Reject

**Metareview:**

This paper demonstrated interesting observations that simple transformations such as a rotation and a translation is enough to fool CNNs. Major concern of the paper is the novelty. Similar ideas have been proposed before by many previous researchers. Other networks trying to address this issue have been proposed. Such as those rotation-invariant neural networks. The grid search attack used in the experiments may be not convincing. Overall, this paper is not ready for publication.